# Exponentiated Strongly Rayleigh Distributions

**Zelda Mariet**
Massachusetts Institute of Technology
zelda@csail.mit.edu

**Suvrit Sra**
Massachusetts Institute of Technology
suvrit@mit.edu

**Stefanie Jegelka**
Massachusetts Institute of Technology
stefje@csail.mit.edu

## Abstract

Strongly Rayleigh (SR) measures are discrete probability distributions over the subsets of a ground set. They enjoy strong *negative dependence* properties, as a result of which they assign higher probability to subsets of *diverse* elements. We introduce in this paper Exponentiated Strongly Rayleigh (ESR) measures, which sharpen (or smoothen) the negative dependence property of SR measures via a single parameter (the exponent) that can be intuitively understood as an inverse temperature. We develop efficient MCMC procedures for approximate sampling from ESRs, and obtain explicit mixing time bounds for two concrete instances: exponentiated versions of Determinantal Point Processes and Dual Volume Sampling. We illustrate some of the potential of ESRs, by applying them to a few machine learning problems; empirical results confirm that beyond their theoretical appeal, ESR-based models hold significant promise for these tasks.

## 1 Introduction

The careful selection of a few items from a large ground set is a crucial component of many machine learning problems. Typically, the selected set of items must fulfill a variety of application specific requirements—e.g., when recommending items to a user, the *quality* of each selected item is important. This quality must be, however, balanced by *diversity* of the selected items to avoid redundancy within recommendations. Notable applications requiring careful consideration of subset diversity include recommender systems, information retrieval, and automatic summarization; more broadly, such concerns are also vital for model design such as model pruning and experimental design.

A flexible approach for such subset selection is to sample from subsets of the ground set using a measure that balances quality with diversity. An effective way to capture diversity is to use *negatively dependent* measures. While such measures have been long studied [41], remarkable recent progress by Borcea et al. [11] has put forth a rich new theory with far-reaching impact. The key concept in Borcea et al.'s theory is that of Strongly Rayleigh (SR) measures, which admit important closure properties (specifically, closure under conditioning on a subset of variables, projection, imposition of external fields, and symmetric homogenization [11, Theorems 4.2, 4.9]) and enjoy the strongest form of negative association. These properties have been instrumental in the resolution of long-standing conjectures in mathematics [9, 35]; in machine learning, their broader impact is only beginning to emerge [5, 31, 33], while an important subclass of SR measures, Determinantal Point Processes (DPPs) has already found numerous applications [22, 29].

A practical challenge in using SR measures is the tuning of diversity versus quality, a task that is application dependent and may require significant effort. The modeling need motivates us to consider a generalization of SR measures that allows for easy tuning of the relative importance given to quality and diversity considerations. Specifically, we introduce the class of *Exponentiated*

*Strongly Rayleigh (ESR)* measures, which are distributions of the form $\nu(S) \propto \mu(S)^p$, where $S$ is a set, $p > 0$ is a parameter and $\mu$ is an SR measure. A power $p > 1$ captures a sharper notion of diversity than $\mu$; conversely, a power $p < 1$ allows for weaker diversity preferences; at the $p = 0$ extreme, $\nu$ is uniform, while for $p \to \infty$, the $\nu$ concentrates at the mode of $\mu$.

ESR measures present an attractive generalization to SR measures, where a single parameter allows an intuitive regulation of desired strength of negative dependence. Interestingly, a few special cases of ESRs have been briefly noted in the literature [22, 29, 49], although only the guise of generalizations to DPPs and without noting any connection to SR measures.

We analyze the negative association properties of ESR measures and derive general-purpose sampling algorithms that we further specialize for important concrete cases. Subsequently, we evaluate the proposed sampling procedures on outlier detection and kernel reconstruction, and show how a class of machine learning problems can benefit from the modeling power of ESR measures.

**Summary of contributions.**    The key contributions of this paper are the following:

– The introduction of Exponentiated SR measures as a flexible generalization of SR measures, allowing for intuitive tuning of subset selection quality/diversity tradeoffs via an exponent $p > 0$.

– A discussion of cases when ESR measures remain SR. Specifically, we show that there exist non-trivial determinantal measures whose ESR versions remain SR for $p$ in a neighborhood of 1.

– The introduction of the notion of $r$-closeness, which quqntifies the suitability of a proposal distribution for MCMC samplers.

– The analysis of MCMC sampling algorithms applied to ESR measures which take advantage of fast-mixing chains for SR measures. We show that the mixing time of the ESR samplers is upper bounded in terms of $r$-closeness; we provide concrete bounds for popular SR measures.

– An empirical evaluation of ESR measures on various machine learning tasks, showing that ESR measures outperform standard SR models on several problems requiring a delicate balance of subset quality and diversity.

## 1.1   Related work

An early work that formally motivates various negative dependence conjectures is [41]. The seminal work [11] provides a response, and outlines a powerful theory of negative dependence via the class of SR measures. The mathematical theory of SR measures, as well as the intimately related theory of multivariate *stable polynomials* has been the subject of significant interest [9, 10, 42]; recently, SR measures were central in the proof of the Kadison-Singer conjecture [35].

Within machine learning, DPPs, which are a subclass of SR measures, have been recognized as a powerful theoretical and practical tool. DPPs assign probability proportional to $\det(L[S])$ to a set $S \in 2^{[n]}$, where $L$ is the so-called DPP-kernel. Their elegance and tractability has helped DPPs find numerous applications, including document and video summarization [15, 34], sensor placement [27], recommender systems [21, 48], object retrieval [1], neural networks [36] and Nyström approximations [32]. More recently, an SR probability measure known as volume sampling [8, 16] or dual volume sampling (DVS) [33, 37] has found some interest. A DVS measure is parametrized by an $m \times n$ matrix $A$ with columns $a_i$; it assigns to a set $S \subseteq [n]$ of size $m$ a probability proportional to $\det(\sum_{i \in S} a_i^\top a_i)$.

Independent of application-specific motivations, two recent results [5, 31] showed that SR measures admit efficient sampling via fast-mixing Markov chains, suggesting SR measures can be tractably applicable to many machine learning problems. Nevertheless, the need to tune the measure to modulate diversity persists. We address this need by passing to the broader class of Exponentiated Strongly Rayleigh measures, whose diversity/quality preference is parametrized by a single exponent.

To our knowledge, there has been no previous discussion of ESR measures as a class. Nonetheless, they can benefit from the abundant existing theory for log-submodular models [19, 20, 25, 43], and isolated special cases have also been discussed in the literature. In particular, Exponentiated DPPs (or E-DPPs) are mentioned in [29, 49], as well as in [22] and [4].

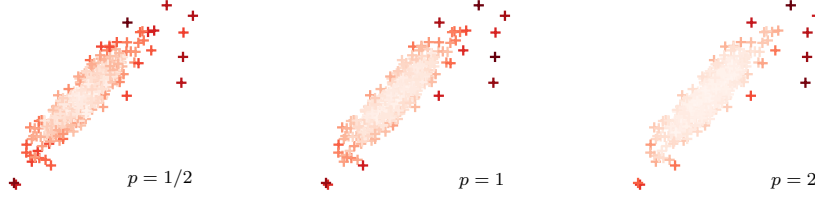

$p = 1/2$     $p = 1$     $p = 2$

Figure 1: Anomaly detection by sampling with an Exponentiated-DPP. 200 samples of size $k = 20$ were drawn from a E-DPP with Gaussian kernel; darker colors indicate higher sampling frequencies. As $p$ increases, the points furthest from the mean accumulate all of the sampling probability mass.

## 2    Exponentiated Strongly Rayleigh measures

In this section, we formally introduce Exponentiated SR measures and analyze their properties within the framework of negative dependence. We use $\mathbb{P}^n$ to denote $n \times n$ Hermitian positive definite matrices, and use $A \succ B$ to denote the usual Löwner order on $\mathbb{P}^n$ matrices[1]. For a matrix $L$, we write $L[S, T]$ the submatrix $[L_{ij}]_{i \in S, j \in T}$, as well as $L[:, S] \triangleq L[[n], S]$ and $L[S, :]$ similarly. We alleviate the notation $L[S, S]$ as $L[S]$.

Recall that for a measure $\mu$ over all subsets of a ground set $\mathcal{Y} \triangleq [n]$, $\mu$'s generating polynomial is the multi-affine function over $\mathbb{C}^n$ defined by

$$P_\mu(z_1, \ldots, z_n) = \sum\nolimits_{S \subseteq \mathcal{Y}} \mu(S) \prod\nolimits_{i \in S} z_i$$

**Definition 1** (Strongly Rayleigh [11]). *A measure $\mu$ over the subsets of $[n] := \{1, \ldots, n\}$ is SR if its generating polynomial $P_\mu \in \mathbb{C}[z_1, \ldots, z_n]$ is real stable, i.e. $P_\mu(z_1, \ldots, z_n) \neq 0$ whenever $\mathfrak{Im}(z_j) > 0$ for $1 \leq j \leq n$.*

In order to calibrate the relative influence of the diversity and quality of a set $S$ on the probability an SR measure assigns to $S$, we introduce the family of *Exponentiated* Strongly Rayleigh measures.

**Definition 2** (Exponentiated SR measure). *A measure $\mu$ over $2^{[n]}$ is Exponentiated Strongly Rayleigh (ESR) if there exists an SR measure $\nu$ over $2^{[n]}$ and a power $p \geq 0$ such that $\mu(S) \propto \nu(S)^p$.*

The parameter $p$ serves to control the quality/diversity tradeoff by sharpening ($p > 1$) or smoothing out ($p < 1$) the variations of the ground SR measure (see Figure 1). A natural question is then to understand how this additional parameter impacts the negatively associated properties of the ESR. Recall that a fundamental property of SR measures lies in the fact that they are negatively associated: for two increasing functions $F, G$ over $2^{[n]}$ that depend on a disjoint set of coordinates, an SR measure $\mu$ verifies the following inequality [11, Theorem 4.9]:

$$\mathbb{E}_\mu[F] \, \mathbb{E}_\mu[G] \geq \mathbb{E}_\mu[FG]. \tag{2.1}$$

Our first result states that the additional modularity enabled by the exponent parameter can break Strong Rayleighness; as a consequence, we have no immediate guarantee that ESRs verify Eq. (2.1).

**Proposition 1.** *There exist ESR measures that are not SR.*

Conversely, some ESR measures remain SR for any $p$: if $\mu$ is a DPP parametrized by a block-diagonal kernel with $2 \times 2$ blocks, $\nu = \alpha \mu^p$ is also a DPP, and so SR and ESR by construction. The next theorem guarantees the existence of non-trivial ESR measures which are also SR.

**Theorem 1.** *There exists $\epsilon > 0$ such that $\forall p \in [1 - \epsilon, 1 + \epsilon], \forall n \in \mathbb{N}$, there exists a non-trivial matrix $L \in \mathbb{P}_n$ such that the E-DPP distribution defined by $\nu(S) \propto \det(L[S, S])^p$ is SR.*

Hence, ESRs are not guaranteed to be SR but may remain so. Due to their log-submodularity, they nonetheless will verify the so-called *negative latice condition* $\mu(S \cap T)\mu(S \cup T) \leq \mu(S)\mu(T)$, and so retain negative dependence properties.

We now show that ESRs nonetheless have a fundamental advantage over standard log-submodular functions: although the intractability of their partition function precludes exact sampling algorithms, their closed form as the exponentiation of an SR measure can be leveraged to take advantage of the recent result [31] on fast-mixing Markov chains for SR measures.

[1]i.e. $A \succ B \iff (A - B) \in \mathbb{P}^n$.

## 3 Sampling from ESR measures

In the general case, the normalization term of an ESR is NP-hard to compute, precluding exact sampling algorithms. In this section, we propose instead two MCMC sampling algorithms whose key idea lies in exploiting the explicit relation ESR measures have to SR measures.

We begin by introducing the notion of $r$-closeness, which serves as a measure of the proximity between to distributions $\mu$ and $\nu$ over subsets. In practice, $r$-closeness will allow us to quantify how close an ESR measure is to being SR, and inform our bounds on mixing time.

**Definition 3** ($r$-closeness). Let $\mu$, $\nu$ be measures over $2^{[n]}$ and let $p \geq 0$. We say that $\nu$ is $r$-*close* to $\mu$ if we have for all $S \subseteq [n]$,

$$\nu(S) \neq 0 \text{ and } \mu(S) \neq 0 \implies r^{-1} \leq \nu(S)/\mu(S) \leq r$$

where we allow $r = \infty$. We additionally write $r(\mu, \nu) = \min\{r \in \mathbb{R} \cup \{\infty\} : \nu \text{ is } r\text{-close to } \mu\}$.

**Remark 1.** If $r(\mu, \nu) < \infty$, $\nu$ is absolutely continuous wrt. $\mu$: $\mu(S) = 0 \implies \nu^p(S) = 0$.

The following result establishes that for any ESR measure $\nu$, there exists an SR measure $\mu$ which is $r$-close to $\nu$ with $r < \infty$. This result is the cornerstone of the sampling algorithms we derive, as we show that we can use an $r$-close SR measure as proposal to efficiently sample from an ESR measure.

**Proposition 2.** *Let $\mu$ be an SR measure over $2^{[n]}$, and define $\nu$ to be the ESR measure such that $\nu(S) \propto \mu(S)^p$. Then*

$$r(\mu, \nu) \leq \max_{S \in \text{supp}(\nu)} \left[ \mu(S)^{-|p-1|} \right] < \infty.$$

In order to sample from an ESR distribution $\nu$, we now generalize existing MCMC algorithms for SR measures; we bound the distance to stationarity of the the chain's current state by comparing it to the distance to stationarity of a similar chain sampling from an SR measure $\mu$, and leveraging the $r$-closeness $r(\mu, \nu)$.

### 3.1 Approximate samplers for ESR measures

Before investigating MCMC samplers, one may first wonder if rejection sampling might be sufficient: sample a set $S$ from a proposal distribution $\mu$, and accept with probability $\nu^p(S)/M\mu(S)$, where $M \geq \max_S \mu(S)/\nu^p(S)$. Unfortunately, the rejection sampling scaling factor $M$ cannot be computed — although it can be bounded by $r(\mu, \nu^p)$ — leading us to prefer MCMC samplers [6].

We begin by analyzing the standard independent Metropolis–Hastings sampler [26, 38], using an SR measure $\mu$ as a proposal: we sample an initial set $S$ from $\mu$ via a fast-mixing Markov chain, then iteratively swap from $S$ to a new set $S'$ with probability

$$\Pr(S \rightarrow S') = \min\left\{1, \frac{\nu(S')\mu(S)}{\nu(S)\mu(S')}\right\}$$

---

**Algorithm 1** Proposal-based sampling

---
**Input:** SR proposal $\mu$, ESR measure $\nu$ and SR measure $\rho$ s.t. $\nu = \alpha\rho^p$
Draw $S \sim \mu$
**while** not mixed **do**
    $S' \sim \mu$
    $S \leftarrow S'$ w.p. $\min\left\{1, \frac{\nu(S')\mu(S)}{\nu(S)\mu(S')}\right\} = \min\left\{1, \frac{\mu(S)}{\mu(S')}\left(\frac{\rho(S')}{\rho(S)}\right)^p\right\}$
**return** $S$

---

Algorithm 1 relies on the fact that we can compute $\nu(S')/\nu(S)$ as $(\rho(S')/\rho(S))^p$: we do not require knowledge of $\nu$'s partition function. This sampling method is valid as soon as $\nu$ is absolutely continuous with regards to the proposal $\mu$; Proposition 2 guaranteed the existence of such measures.

If the ESR measure $\nu$ is $k$-homogeneous (i.e. $\nu$ assigns a non-zero probability only to sets of size $k$), we can instead sample from $\nu$ via Algorithm 2: we randomly sample $S \subseteq [n]$ and switch an element $u \in S$ for $v \notin S$ if this improves the probability of $S$.

---

**Algorithm 2** Swap-chain sampling

---

**Input:** $k$-homogeneous ESR measure $\nu$ s.t. $\nu = \alpha\rho^p$ with $\rho$ SR
Sample $S \sim \text{Unif}(N)$ such that $|S| = k$
**while** not mixed **do**
    Sample $u, v \in (S \times [N] \setminus S)$ uniformly at random
    $S \leftarrow S \cup \{u\} \setminus \{v\}$ w.p. $\min\left\{1, \frac{\nu(S\cup\{u\}\setminus\{v\})}{\nu(S)}\right\} = \min\left\{1, \left(\frac{\rho(S\cup\{u\}\setminus\{v\})}{\rho(S)}\right)^p\right\}$
**return** $S$

---

The key to extending Algorithm 2 to non-homogeneous ESR measures is similar to the approach taken by Li et al. [31] for SR measures, and relies on leveraging the symmetric homogenization $\nu_{\text{sh}}$ of $\nu$ over $2^{[2n]}$ defined by

$$\nu_{\text{sh}} : S \in 2^{[2n]} \to \begin{cases} \nu(S \cap [n])\binom{n}{|S\cap[n]|}^{-1} & \text{if } |S| = n \\ 0 & \text{if } |S| \neq n \end{cases}$$

If $\nu \propto \mu^p$, $\nu_{\text{sh}}$ is absolutely continuous with regards to $\mu_{\text{sh}}$. A simple calculation further shows that $r(\mu_{\text{sh}}, \mu_{\text{sh}}) = r(\mu, \nu)$, and so to sample $S$ from $\nu$, it suffices to sample $T$ of size $n$ from $\nu_{\text{sh}}$ using Algorithm 2, and then output $S = T \cap [n]$.

Hence, although we cannot in the general case sample from an ESR measure exactly (unlike many SR measures), being able to evaluate an ESR measure's unnormalized density function allows us to leverage MCMC algorithms for approximate sampling. We now focus on bounding the mixing times of these algorithms.

### 3.2 Bounds on mixing time for the proposal and swapchain algorithms

Writing $\nu'_{t,S}$ the distribution generated by a Markov chain sampler after $t$ iterations and initialization set $S$, the mixing time $\tau_S(\epsilon)$ measures the number of required iterations of the Markov chain so that $\nu'_{t,S}$ is close enough (in total variational distance) to the true ESR measure $\nu$:

$$\tau_S(\epsilon) \triangleq \min\{t : \|\nu'_{t,S} - \nu\|_{\text{TV}} \leq \epsilon\}$$

It is easy to see from the above equation that the mixing time of a chain depends on how close the distribution generating the initialization set $S$ is to the target distribution $\mu$. We now show this explicitly for the two algorithms derived above, obtaining bounds on $\tau_S$ that directly depend on the $r$-closeness of the target ESR measure $\nu$ and an SR measure $\mu$.

For Algorithm 1, the mixing time explicitly depends on the quality of the proposal distribution.

**Theorem 2** (Alg. 1 mixing time). *Let $\mu, \nu$ be measures over $2^{[n]}$ such that $\mu$ is SR and $\nu$ is ESR. Sampling from $\nu$ via Alg. 1 with $\mu$ as a proposal distribution has a mixing time $\tau(\epsilon)$ such that*

$$\tau_S(\epsilon) \leq 2r(\mu, \nu^p)\log\frac{1}{\epsilon}.$$

For the swapchain algorithm (Alg. 2), we derive a bound on the mixing time by comparing to a result by [5] which shows fast sampling for SR distributions over subsets of a fixed size.

**Theorem 3** ( Alg. 2 mixing time). *Let $\nu$ be a $k$-homogeneous ESR measure over $2^{[n]}$. The mixing time for Alg. 2 with initialization $S$ is bounded in expectation by*

$$\tau_S(\epsilon) \leq \inf_{\mu\in SR} 2nk\, r(\mu,\nu)^2 \log\frac{1}{\epsilon\nu(S)}$$

The above bound depends on the *closest* SR distribution to the target measure $\nu$. Combined with Prop. 2, Thm. 3 provides a simple upper bound to the mixing time of the swapchain algorithm.

**Corollary 1** (Non-homogeneous swapchain mixing time). *Let $\nu$ be a non-homogeneous ESR measure over $2^{[n]}$. The mixing time for the generalized swapchain sampler to sample from $\nu$ with initialization $S \subseteq [2n]$ is bounded in expectation by*

$$\tau_S(\epsilon) \leq \inf_{\mu\in\text{SR}} 4n^2\, r(\mu,\nu)^2 \log\frac{1}{\epsilon\nu_{\text{sh}}(S)}$$

As a Markov chain's applicability closely depends on its mixing time, a crucial task in sampling from ESR measures lies in finding an $r$-close SR distribution with small $r$.

### 3.3 Specific bounds for $r$-closeness

We now derive explicit mixing time bounds for ESR measures $\nu$ generated by two popular classes of SR measures: DPPs, in their usual form as well as their $k$-homogeneous form ($k$-DPPs), and Dual Volume Sampling (DVS). As Theorem 2 and Theorem 3 provide mixing time bounds that depend explicitly on $r(\mu, \nu)$, this section focuses on upper bounding $r(\mu, \nu)$. To the extent of our knowledge, the results below are the first for either of these two classes of ESR distributions.

**Theorem 4** (E-DVS closeness bounds). *Let $n \geq k \geq m$ and let $X \in \mathbb{R}^{m \times n}$ be a maximal-rank matrix. Let $\mu$ be the Dual-Volume Sampling distribution over $2^{[n]}$ for sets of size $k$:*

$$\mu : S \subseteq [n] \to \begin{cases} 0 & \text{if } |S| \neq k \\ \mu(S) \propto \det(X[:, S]X[:, S]^\top) & \text{if } |S| = k \end{cases}$$

*Let $p > 0$ and $\nu$ be the ESR measure induced by $\mu$ and $p$; let $\mathrm{MinVol}(X, S)$ be the smallest non-zero minor of degree $m$ of $X[:, S]$. Then*

$$r(\mu, \nu) \leq \binom{n - m}{k - m}^{|1-p|} \binom{k}{m}^{-|1-p|} \det(XX^\top)^{|1-p|} \mathrm{MinVol}(X, S)^{-2|1-p|}$$

**Theorem 5** (E-DPP closeness bound). *Let $\mu$ be the distribution induced by a DPP with kernel $L \succeq 0$ and $\nu$ be the E-DPP such that $\nu(S) \propto \det(L[S])^p$. Let $\lambda_1 \leq \cdots \leq \lambda_n$ be the ordered eigenvalues of $L$. Then,*

$$r(\mu, \nu^p) \leq \prod_{i=1}^{n} (1 + \lambda_i)^{|1-p|} \prod_{\lambda_i < 1} \lambda_i^{-|1-p|}.$$

**Theorem 6** (E-$k$-DPP closeness bound). *Let $\mu$ be the distribution over $2^{[n]}$ induced by a $k$-DPP ($k \leq n$) with kernel $L$, and let $\nu$ be the induced ESR measure with power $p > 0$. Then*

$$r(\mu, \nu) \leq e_k(\lambda_1, \ldots, \lambda_n)^{|1-p|} \prod_{i=1}^{k} \lambda_i^{-|1-p|}.$$

*where $e_k$ the $k$-th elementary symmetric polynomial.*

One easily shows that the values $r(\mu, \nu)$ we derive above for ($k$-) DPPs are loosely lower-bounded by $\kappa^{|1-p|}$, where $\kappa$ is the condition number of the kernel matrix $L$. However, it is possible to obtain a closer SR distribution to $\nu \propto \det(L)^p$ than the baseline choice of the DPP with kernel $L$: indeed, as $L$ is positive semi-definite, we can also consider a DPP parametrized by kernel $L^p$.

For the rest of this section, we define $\mu$ as the SR measure corresponding to the DPP with kernel $L^p$: $\mu(S) = \det(L^p[S]) / \det(I + L^p)$, and $\nu$ as the ESR measure such that $\mu(S) \propto \det(L[S])^p$. Note that $\nu$ remains absolutely continuous with regard to $\mu$. In this setting, upper bounding $r(\mu, \nu)$ proves to be significantly more difficult, and is the focus of the remainder of this section. We first recall a useful expansion of the determinant of principal submatrices, fundamental to deriving the bounds below and potentially of more general interest.

**Lemma 1** (Shirai and Takahashi [44, Lemma 2.9]). *Let $H$ be an $n \times n$ Hermitian matrix with eigenvalues $\lambda_1, \ldots, \lambda_n$. There exists a $2^n \times 2^n$ symmetric doubly stochastic matrix $Q = [Q_{SJ}]$ indexed by subsets $S, J$ of $[n]$ such that*

$$\det(H[S]) = \sum_{J \subseteq [n], |J| = |S|} Q_{SJ} \prod_{i \in J} \lambda_i.$$

*$Q$ can be chosen to depend only on the eigenvectors of $H$ and to satisfy $Q_{SJ} = 0$ for $|S| \neq |J|$.*

The above lemma allows us to bound $\frac{\det(L^p[S])}{\det(L[S]^p)}$ in terms of the *generalized condition number* of $L$.

**Definition 4** (Generalized condition number). Given a matrix $L \in \mathbb{P}^n$ with eigenvalues $\lambda_1, \ldots, \lambda_n$, we define its generalized condition number of order $k$ as

$$\kappa_k = (\lambda_1 \cdots \lambda_k)(\lambda_n \cdots \lambda_{n-k})^{-1}.$$

Note that $\kappa_k$ is the usual condition number of the $k$-th exterior power $L^{\wedge k}$ (in particular $\kappa_k \geq \kappa^k$).

Given the generalized conditioned number, Lemma 1 combined with the power-mean inequality [45] (see App. D) suffices to bound the gap between volumes generated by E-DPPs and DPPs:

**Theorem 7.** *Let $\mu$ be the distribution induced by a* DPP *with kernel $L^p$, and $\nu$ be the corresponding* E-DPP *such that $\nu \propto \det(L[S])^p$. Then $r(\mu,\nu) \le r(\kappa_{\lfloor n/2 \rfloor}, p)$ where $r(\kappa, p)$ is defined by*

$$r(\kappa, p) = \begin{cases} \left(\frac{p(\kappa-1)}{\kappa^p-1}\right)^p \left(\frac{(1-p)(\kappa-1)}{\kappa-\kappa^p}\right)^{1-p} & \text{for } 0 < p < 1 \\ \left(\frac{\kappa^p-1}{p(\kappa-1)}\right)^p \left(\frac{(p-1)(\kappa-1)}{\kappa^p-\kappa}\right)^{p-1} & \text{for } p > 1 \end{cases}$$

**Corollary 2.** *Let $\mu$ be the distribution induced by a $k$-DPP with kernel $L^p$, and $\nu$ be the corresponding ESR measure such that $\nu(S) \propto \det(L[S])^p$. Then $r(\mu,\nu) \le r(\kappa_k, p)$.*

As shown in Figure 4 (App. D), the upper bound 7 grows slower than $\kappa$: this shows that the $\mu(S) \propto \det(L^p[S])$ is a closer SR distribution to an E-DPP with kernel $L$ than the E-DPP's generating SR distribution, and leads to finer mixing time bounds.

Note that the per-iteration complexity of both algorithms must also be taken into account when choosing a sampling procedure: for E-DPPs, despite Alg. 1's smaller mixing time, Alg. 2 is more efficient in cases when $n$ large due to the comparative costs of each sampling round.

## 4 Experiments

To evaluate the empirical applications of ESR measures, we evaluate E-DPPs (DPPs are by far the most popular SR measure in machine learning) on a variety of machine learning task. In all cases where we use the proposal MCMC sampler (Alg. 1), we use the DPP with kernel $L^p$ as a proposal.

### 4.1 Evaluating mixing time

We begin our experiments by empirically evaluating the mixing time of both algorithms. We measure mixing using the classical Potential Scale Reduction Factor (PSRF) metric [13]. As the PSRF converges to 1, the chain mixes. In the following experiments, we report the mixing time (number of iterations) necessary to reach a PSRF of 1.05, as well as the runtime (in seconds) to convergence, averaged over 5 iterations; we use matrices with a fixed $\kappa^k$ across all mixing time experiments.

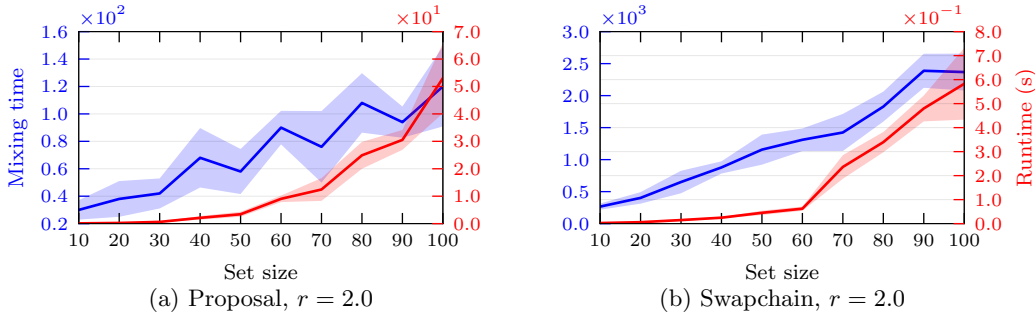

(a) Proposal, $r = 2.0$      (b) Swapchain, $r = 2.0$

Figure 2: Mixing and sampling time for E-$k$-DPPs as a function of the set size $k$. In both cases, the mixing time grows linearly with $k$; although the mixing time for the proposal algorithm is an order of magnitude smaller than for the swapchain algorithm, the latter samples faster due to the per-iteration cost of each transition step.

The mixing time for proposal-based sampling is an order of magnitude smaller than swap-chain sampling; this is in line with the bounds we provide in Theorems 2 and 3. However, this does not translate into faster runtimes: indeed, the per-iteration complexity of proposal-based sampling is significantly higher than for the swapchain algorithm, as Alg. 1 samples from a DPP at each iteration. The evolution of mixing and wall clock times as a function of $N$ is provided in Appendix E.

### 4.2 Anomaly detection

We now focus on applications for E-DPPs; we begin by evaluating the use of E-DPPs for outlier detection. As increasing $p$ hightens the model's sensitivity to diversity, we expect $p > 1$ to provide better outlier detection. To our knowledge, this is the first application of DPPs to outlier detection, and so our goal for this experiment is not to improve upon state-of-the-art results, but to compare the performance of (E-)DPPs for various values of $p$ to standard outlier detection algorithms.

Experimentally, we detect an outlier via the following approach: given a dataset of $n$ points and an E-DPP with an RBF kernel built from the data (bandwidth $\beta = 100$), we sample $\frac{n}{5}$ subsets of size 50

and report as outliers points that appear at least $n\varepsilon$ times, where $\varepsilon$ is a tunable parameter (hence, if we were doing uniform sampling, each point in the dataset would be sampled on average 10 times).

We detect outliers on three public datasets: the UCI Breast Cancer Wisconsin dataset [46] modified as in [24, 28] as well as the Letter and Speech datasets fom [39]. We also report the performance of a selection of standard outlier detection algorithms whose reported performance in [24] is competitive with other outlier detection algorithms: Local Outlier Factor (LOF) [12], $k$-Nearest Neighbor ($k$-NN) [7], Histogram-based Outlier Score (HBOS) [23], Local Outlier Probability (LoOP) [28] and unweighted Cluster-Based Local Outlier Factor (uCBLOF) [3, 24].

| $p$ | 0.5 | 1 | 2 | LOF* | $k$-NN* | HBOS* | LoOP* | uCBLOF* |
|---|---|---|---|---|---|---|---|---|
| Cancer | $0.952 \pm 0.018$ | $0.962 \pm 0.004$ | $0.965 \pm 0.001$ | $0.982 \pm 0.002$ | $0.979 \pm 0.001$ | $0.983 \pm 0.002$ | $0.973 \pm 0.012$ | $0.950 \pm 0.039$ |
| Letter | $0.780 \pm 0.013$ | $0.820 \pm 0.003$ | $0.847 \pm 0.002$ | $0.867 \pm 0.027$ | $0.872 \pm 0.018$ | $0.622 \pm 0.007$ | $0.907 \pm 0.008$ | $0.819 \pm 0.023$ |
| Speech | $0.455 \pm 0.007$ | $0.439 \pm 0.011$ | $0.445 \pm 0.002$ | $0.504 \pm 0.022$ | $0.497 \pm 0.010$ | $0.471 \pm 0.003$ | $0.535 \pm 0.034$ | $0.469 \pm 0.003$ |

Table 1: AUC (mean + standard deviation) for E-DPPs and standard outlier detection algorithms. As expected, we see that a higher exponent leads to a stronger preference for diversity and hence a better outlier detection scheme. Only LoOP and LOF consistently outperform E-DPPs.

Results are reported in Table 1; as expected, we see that larger values of $p$ (in this case, $p = 2$) are more sensitive to outliers, and provide better models for outlier detection.

### 4.3 E-DPPs for the Nyström method

As a more standard application of DPPs, we now investigate the use of E-DPPs for kernel reconstruction via the Nyström method [40, 47]. Given a large kernel $K$, the Nyström method selects a subset $C$ of columns ("landmarks") of $K$ and approximates $K$ as $K[:, C]K^\dagger[C, C]K[C, :]$. Unsurprisingly, DPPs have successfully been applied to the landmark selection for the Nyström approach [2, 30]. We show here that E-DPPs further improve upon the recent results of [30] for kernel reconstruction.

We apply Kernel Ridge Regression to 3 regression datasets: Ailerons, Bank32NH, and Machine CPU[2]. We subsample 4,000 points from each dataset (3,000 training and 1,000 test) and use an RBF kernel and choose the bandwidth $\beta$ and regularization parameter $\lambda$ for each dataset by 10-fold cross-validation. Results are averaged over 3 random subsets of data, using the swapchain sampler initialized with $k$-means++ and run for 3000 iterations.

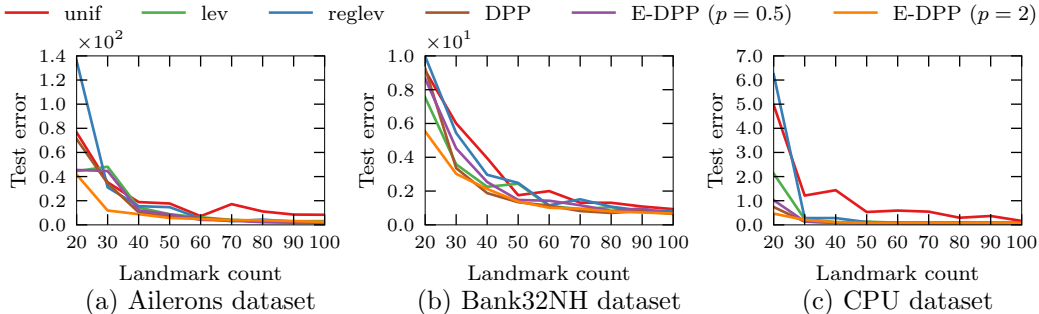

(a) Ailerons dataset  (b) Bank32NH dataset  (c) CPU dataset

Figure 3: Prediction error on regression datasets; we compare various E-DPP models to uniform sampling ("unif") as well as leverage and regularized leveraged sampling ("lev" and "reglev"). On all datasets, the E-DPPs achieve the lowest error, with the largest exponent $p = 2$ performing markedly better than other methods.

We evaluate the quality of the sampler via the prediction error on the held-out test set. Figure 3 reports the results. Consistently across all datasets, $p = 2$ outperforms all other samplers in terms of the prediction error, in particular when only sampling a few landmarks. Interestingly, we also see that the reconstruction error tends to be smaller when $p = \frac{1}{2}$ (see Appendix F).

## 5 Conclusion and extensions

Many machine learning problems have been shown to benefit from the negative dependence properties of Strongly Rayleigh measures: measures based on elementary symmetric polynomials – including (dual) volume sampling – have been applied to experimental design; DPPs have been applied

successfully to fields ranging from automatic summarization to minibatch selection and neural network pruning. However, tuning the strength of the quality/diversity tradeoff of SR measures requires significant effort.

We introduced Exponentiated Strongly Rayleigh measures, an extension of Strongly Rayleigh measures which augment standard SR measures with an exponent $p$, allowing for straightforward tuning of the the quality-diversity trade-off of SR distributions. Intuitively, $p$ controls how much priority should be given to diversity requirements. We show that although ESR measures do not necessarily remain SR, but certain distributions lie at the intersection of both classes.

Despite their intractable partition function, ESR measures can leverage existing fast-mixing Markov chains for SR measures, enabling finer bounds than those obtained for the broader class of log-submodular models. We derive general-purpose mixing bounds based on the distance from the target distribution $\nu$ to an SR distribution $\mu$; we then show that these bounds can be further improved by specifying a carefully calibrated SR proposal distribution $\mu$, as is the case for Exponentiated DPPs.

We verified empirically that ESR measures and the algorithms we derive are valuable modeling tools for machine learning tasks, such as outlier detection and kernel reconstruction. Finally, let us note that there remain several theoretical and practical open questions regarding ESR measures; in particular, we believe that further specifying the class of ESR measures that also remain SR may provide valuable insight into the study of negatively associated measures.

Finally, one easily verifies that given $\mu$ SR and a collection of *i.i.d.* subsets $\mathcal{S} = \{S_1, \ldots, S_m\}$, the MLE problem that finds the best $p > 0$ to model $\mathcal{S}$ as being sampled from an ESR $\nu \propto \mu^p$ is *convex*:

$$\operatorname{argmax}_{p>0} \frac{p}{m} \sum\nolimits_{k=1}^{m} \log \mu(S_i) - \log \Big( \sum\nolimits_{S \subseteq [n]} \mu(S)^p \Big). \tag{5.1}$$

As such, standard convex optimization algorithms can be leveraged to select $p$, potentially after leaning a parametrization of $\mu$.

**Acknowledgements.** This work is in part supported by NSF CAREER award 1553284, NSF-BIGDATA award 1741341, and by The Defense Advanced Research Projects Agency (grant number YFA17 N66001-17-1-4039). The views, opinions, and/or findings contained in this article are those of the author and should not be interpreted as representing the official views or policies, either expressed or implied, of the Defense Advanced Research Projects Agency or the Department of Defense.

## Footnotes

[2] http://www.dcc.fc.up.pt/~ltorgo/Regression/DataSets.html

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
