[Supplementary Material]

## A   Negative association properties of ESR measures

**Proposition 1.** There exist ESR measures that are not SR.

*Proof.* Recall [11, Thm. 4.1] that a real bivariate affine polynomial $p(x, y)$ is stable if and only if

$$\partial_x p \partial_y p - p \, \partial_{xy} p \geq 0.$$

For an E-DPP with a kernel $L \in \mathbb{R}^{2\times 2}$, this is exactly $l_{11}^p l_{22}^p \geq \det(L)^p$, which is clearly true for all $p \geq 0$ as $L$ must be positive semi-definite.

However, stability does not hold in general. To obtain a counterexample, consider the generating polynomial $p(x, y, z)$ for an E-DPP of dimension 3 (writing $d = \det(L)$ as a shorthand):

$$p(x, y, z) := d^p + l_{11}^p yz + l_{22}^p xz + l_{33}^p xy + \det(L[1, 2])^p z + \det(L[1, 3])^p y + \det(L[2, 3])^p x + xyz.$$

If $p(x, y, z)$ is SR, $p$ must be stable through conditioning [11, Theorem 4.1]; hence, $p(x, y, 1)$ must also be stable. Writing $d_{ij} = \det(L[i, j])$, this requires that

$$p(x, y, 1) = d^p + d_{12}^p + (d_{22}^p + d_{23}^p)x + (d_{11}^p + d_{13}^p)y + (d_{33}^p + 1)xy$$

be stable. Since $p(x, y, 1)$ is a real bivariate affine polynomial, we must then have

$$(d_{22}^p + d_{23}^p)(d_{11}^p + d_{13}^p) \geq (d^p + d_{12}^p)(d_{33}^p + 1).$$

Finally, one can verify that this last inequality is easily violated for several choices of (non-diagonal) positive semi-definite matrices. $\square$

**Theorem 1.** There exists $\epsilon > 0$ such that for any $p \in [1 - \epsilon, 1 + \epsilon]$ and $n \in \mathbb{N}$, the set $\{L \in \mathbb{R}^{n\times n} : \text{E-DPP}(L, p) \text{ is SR}\}$ is strictly greater than the set of block-diagonal matrices with $2 \times 2$ blocks.

*Proof.* We write $d_S^i = \det(L[S \cup \{i\}))^p$ and when possible $d_{ijk} = \det(L[\{i, j, k\}])^p$.

Let $L \succeq 0 \in \mathbb{R}^{n\times n}$. The associated E-DPP is SR if and only if $P_{ij}(z) \geq 0$ where $P_{ij}(z)$ is defined for any $1 \leq i \neq j \leq n$ as

$$P_r(z) = \frac{\partial f}{\partial z_i}(z)\frac{\partial f}{\partial z_j}(z) - f(z)\frac{\partial^2 f}{\partial z_i \partial z_j}(z)$$

$$= \Big( \sum_{S\in\mathcal{Y}'} d_S^i z^S \Big)\Big( \sum_{S\in\mathcal{Y}'} d_S^j z^S \Big) - \Big( \sum_{S\in\mathcal{Y}'} d_S^{ij} z^S \Big)\Big( \sum_{S\in\mathcal{Y}'} d_S z^S \Big).$$

where we write $\mathcal{Y}' = \{S \in [n], i \notin S, j \notin S\}$ and $z = (z_1, \ldots, z_n)$ where components $z_i$ and $z_j$ are removed. Now, choose $k \in [n]\backslash\{i, j\}$, and write $\tilde{z}$ the vector $z$ without component $z_k$. Write also $\mathcal{Y}'_k = \{S \in \mathcal{Y}', k \notin S\}$. Then,

$$P_{ij}(z) = (\sum_{k\notin S} d_S^i \tilde{z}^S + z_k \sum_{k\in S} d_S^i \tilde{z}^S)(\sum_{k\notin S} d_S^j \tilde{z}^S + z_k \sum_{k\in S} d_S^j \tilde{z}^S)$$

$$- (\sum_{k\notin S} d_S^{ij} \tilde{z}^S + z_k \sum_{k\in S} d_S^{ij} \tilde{z}^S)(\sum_{k\notin S} d_S \tilde{z}^S + z_k \sum_{k\in S} d_S \tilde{z}^S)$$

$$= \Big( \sum_{S\in\mathcal{Y}'_k} d_S^{ik} \tilde{z}^S \sum_{S\in\mathcal{Y}'_k} d_S^{jk} \tilde{z}^S - \sum_{S\in\mathcal{Y}'_k} d_S^{ijk} \tilde{z}^S \sum_{S\in\mathcal{Y}'_k} d_S^k \tilde{z}^S \Big) z_k^2$$

$$+ \Big( \sum_{S\in\mathcal{Y}'_k} d_S^{ik} \tilde{z}^S \sum_{S\in\mathcal{Y}'_k} d_S^j \tilde{z}^S + \sum_{S\in\mathcal{Y}'_k} d_S^i \tilde{z}^S \sum_{S\in\mathcal{Y}'_k} d_S^{jk} \tilde{z}^S$$

$$- \sum_{S\in\mathcal{Y}'_k} d_S^{ijk} \tilde{z}^S \sum_{S\in\mathcal{Y}'_k} d_S \tilde{z}^S - \sum_{S\in\mathcal{Y}'_k} d_S^{ij} \tilde{z}^S \sum_{S\in\mathcal{Y}'_k} d_S^k \tilde{z}^S \Big) z_k$$

$$+ \Big( \sum_{S\in\mathcal{Y}'_k} d_S^i \tilde{z}^S \sum_{S\in\mathcal{Y}'_k} d_S^j \tilde{z}^S - \sum_{S\in\mathcal{Y}'_k} d_S^{ij} \tilde{z}^S \sum_{S\in\mathcal{Y}'_k} d_S \tilde{z}^S \Big)$$

$$= A z_k^2 + B z_k + C$$

Hence, $L$ yields a SR E-DPP measure if and only if the following inequalities hold for all $i, j, k$:

$$(B/2)^2 \leq AC; \quad A \geq 0; \quad C \geq 0. \tag{A.1}$$

When $n = 3$, Eq. (A.1) reduces to the following arithmetic-geometric inequality, as $\mathcal{Y}'_k = \{\emptyset\}$:

$$\left( \frac{d_i d_{jk} + d_j d_{ik} - d_{ij} d_k - d_{ijk}}{2} \right)^2 \leq (d_i d_j - d_{ij})(d_{jk} d_{ik} - d_k d_{ijk}) \tag{A.2}$$

One can easily obtain 3 positive semi-definite matrices $L$ which verify Eq. (A.2) strictly for $p = 1$; in particular, by continuity, there exists $\epsilon > 0$ such that the E-DPP generated by the kernel $L$ and power $p \in [1 - \epsilon, 1 + \epsilon]$ will still verify Eq. (A.2).

Then, as a block-diagonal matrix such that each diagonal block yields SR E-DPPs also yields a SR E-DPP, we can thus generate block-diagonal matrices of any size $n$ such that the blocks are either $L$ or $2 \times 2$ matrices, which all yield SR E-DPPs for $p \in [1 - \epsilon, 1 + \epsilon]$. $\quad\square$

# B  $r$-closeness

**Proposition 2.** Let $\mu$ be an SR measure over $2^{[n]}$, and define $\nu$ to be the ESR measure such that $\mu(S) = \alpha \mu(S)^p$ for a given $\alpha \in \mathbb{R}$. Then

$$r(\mu, \nu) \leq \max_{S \in \text{supp}(\nu)} \left[ \mu(S)^{-|p-1|} \right] < \infty.$$

*Proof.* Let $\mu, \nu$ be as in the proposition statement, and consider $S \in \text{supp}(\nu)$: $\nu(S) > 0$. Recall that $\sum_T \nu(T) = 1$.

$$\frac{\nu(S)}{\mu(S)} = \frac{\mu(S)^p}{\mu(S) \sum_T \mu(T)^p}$$

If $p \leq 1$, we have $\sum_T \mu(T)^p \geq 1$ and $\mu(S)^p \geq \mu(S)$, and so

$$(\min_S \mu(S))^{p-1} \leq \frac{\sum_T \mu(T)}{\sum_T \mu(T)^p} = \frac{1}{\sum_T \mu(T)^p} \leq \frac{\mu(S)^p}{\mu(S) \sum_T \mu(T)^p} \leq \mu(S)^{p-1} \leq \frac{1}{\min_S \mu(S)^{1-p}}$$

where the left inequality is obtained by noticing that $\frac{a+b}{c+d} \geq \min\left( \frac{a}{c}, \frac{b}{d} \right)$.

Similarly, for $p \geq 1$, we have $\sum_T \mu(T)^p \leq 1$ and $\mu(T)^p \leq \mu(T)$, and so

$$\min_S \mu(S)^{p-1} \leq \frac{\mu(S)^p}{\mu(S)} \leq \frac{\mu(S)^p}{\mu(S) \sum_T \mu(T)^p} \leq \frac{\sum_T \mu(T)}{\sum_T \mu(T)^p} \leq \max \mu(S)^{1-p} = \frac{1}{\min \mu(S)^{p-1}}.$$

$\quad\square$

# C  Bounds on mixing times

**Theorem 2.** Let $\mu, \nu$ be measures over $2^{[n]}$ such that $\mu$ is SR and $\nu$ is ESR. Sampling from $\nu$ via Alg. 1 with $\mu$ as a proposal distribution has a mixing time $\tau(\epsilon)$ such that

$$\tau_S(\epsilon) \leq 2r(\mu, \nu^p) \log \frac{1}{\epsilon}.$$

*Proof.* Alg. 1 has a state-independent proposal distribution $\mu$, and hence its mixing time is governed by a ratio of probabilities: Cai [14] showed that, after $t$ iterations,

$$\max_{S,T} d_{\text{TV}}(\nu_{(t)}(\cdot \mid S), \nu_{(t)}(\cdot \mid T)) = \left( 1 - \frac{1}{\max_U \nu(U)/\mu(U)} \right)^t$$

where $\nu_{(t)}(U \mid S)$ is the probability of being in state $U$ after $t$ iterations when starting from set $S$, and $d_{\text{TV}}$ is the total variation distance.

Hence, following [14, Cor.1], we obtain $\tau_S(\epsilon) \leq 2 \max_U \frac{\nu(U)}{\mu(U)} \log \frac{1}{\epsilon} \leq 2r(\mu, \nu^p) \log \frac{1}{\epsilon}$. $\quad\square$

**Theorem 3.** Let $\nu$ be a $k$-homogeneous ESR measure over $2^{[n]}$. The mixing time for Alg. 2 with initialization $S$ is bounded in expectation by

$$\tau_S(\epsilon) \leq \inf_{\mu \in \mathrm{SR}} 2nk\, r(\mu, \nu)^2 \log \tfrac{1}{\epsilon \nu(S)}$$

*Proof.* This bound is based on a comparison method [17], and relates the mixing time to the spectral gap. Let $1 = \mu_1 \geq \mu_2 \geq \ldots \geq -1$ be the eigenvalues of the state transition matrix of the chain. The *spectral gap* is $\gamma = 1 - \max\{|\mu|; \mu \text{ is an eigenvalue and } \mu \neq 1\}$. $\gamma$ directly translates into a bound on the mixing time [18]:

$$\tau_S(\gamma) \leq \frac{1}{\gamma} \log\left(\tfrac{1}{\epsilon \nu(S)}\right).$$

The comparison method yields a bound on $\gamma$ if we know a bound on $\tilde{\gamma}$ for a related chain with stationary distribution $\mu$. Specifying [17, Thm 2.1] to this case yields $\gamma \geq \tilde{\gamma}\alpha_1/\alpha_2$, where

$$\alpha_1 = \min_S \frac{\mu(S)}{\nu(S)} \geq \frac{1}{r(p)}$$

$$\alpha_2 = \max_{T,U} \frac{\mu(T)}{\nu(T)} \frac{\min\{1, \mu(U)/\mu(T)\}}{\min\{1, \nu(U)/\nu(T)\}} \leq \max_T \frac{\mu(T)}{\nu(T)} \leq r(\mu, \nu)$$

Anari et al. [5] show that $\tilde{\gamma} \geq \frac{1}{2nk}$. Hence, we obtain $\tau_S(\gamma) \leq 2nk \cdot r(\mu, \nu)^2 \cdot \log \tfrac{1}{\epsilon \nu(S)}$. $\qquad\square$

# D  Bounds for E-DPPs with $L^p$-kernel proposal

We require the following power-mean inequality:

**Theorem 8** (Specht [45])**.** *Let* $x_i > 0$ *and* $w_i \geq 0$ *for* $1 \leq i \leq N$ *such that* $\sum_i w_i = 1$. *Let* $p < q \in \mathbb{R}$ *such that* $pq \neq 0$. *Then, letting* $\kappa = \frac{\max x_i}{\min x_i}$,

$$1 \leq \frac{M_q(\boldsymbol{w}; \boldsymbol{x})}{M_p(\boldsymbol{w}; \boldsymbol{x})} \leq \left(\frac{q-p}{q} \frac{\kappa^q - 1}{\kappa^q - \kappa^p}\right)^{\frac{1}{p}} \left(\frac{p}{q-p} \frac{\kappa^q - \kappa^p}{\kappa^p - 1}\right)^{\frac{1}{q}}.$$

*where the* power mean $M_p(\boldsymbol{w}; \boldsymbol{x})$ *is defined as*

$$M_p(\boldsymbol{w}; \boldsymbol{x}) := \left(\sum_{i=1}^N w_i x_i^p\right)^{\frac{1}{p}}.$$

**Theorem 9.** *Let* $L \in \mathbb{P}^n$ *be a positive definite matrix and* $S \subseteq [n]$. *Then,*

$$\det(L[S])^p \geq \det(L^p[S]), \qquad 0 \leq p \leq 1,$$
$$\det(L[S])^p \leq \det(L^p[S]), \qquad p \geq 1.$$

*Proof.* From Lemma 1, there exists a vector $\boldsymbol{w}$ in the probability simplex, of size $\binom{n}{|S|}$, such that

$$\det(L[S]) = \sum_{J \subseteq [n], |J| = |S|} w_J \prod_{i \in J} \lambda_i.$$

Since $t \to t^p$ is convex for $p \geq 1$, Jensen's inequality shows that

$$\det(L[S])^p \leq \sum_{J \subseteq [n], |J| = |S|} w_J \prod_{i \in J} \lambda_i^p = \det(L^p[S]),$$

where the latter equality follows due to $L$ and $L^p$ sharing the same eigenbasis. The same reasoning for $p < 1$ gives the other side of the inequality. $\qquad\square$

**Theorem 7.** *Let* $\mu$ *be the distribution induced by a* DPP *with kernel* $L^p$, *and* $\nu$ *be the corresponding* E-DPP *such that* $\nu(S) \triangleq \det(L[S])^p/Z_p$. *Then* $r(\mu, \nu) \leq r(\kappa_{\lfloor n/2 \rfloor}, p)$ *where* $r(\kappa, p)$ *is defined by*

$$r(\kappa, p) = \begin{cases} \left(\frac{p(\kappa-1)}{\kappa^p - 1}\right)^p \left(\frac{(1-p)(\kappa-1)}{\kappa - \kappa^p}\right)^{1-p} & \text{for } 0 < p < 1 \\ \left(\frac{\kappa^p - 1}{p(\kappa-1)}\right)^p \left(\frac{(p-1)(\kappa-1)}{\kappa^p - \kappa}\right)^{p-1} & \text{for } p > 1 \end{cases}$$

*Proof.* We show the result for general DPPs; the result for $k$-DPPs follows the same exact reasoning. For $0 < p < 1$, it follows from Thm. 9 that $\det(L[S])^p \geq \det(L^p[S])$. Hence,

$$Z_p = \sum_{S \subseteq [n]} \det(L[S])^p \geq \sum_{S \subseteq [n]} \det(L^p[S]) = \det(I + L^p),$$

which entails $\frac{\det(I+L^p)}{Z_p} \leq 1$ whereby it remains to bound $\frac{\det(L[S])^p}{\det(L^p[S])}$.

Let $S \subseteq [n]$ of size $k$, and let $\lambda$ be the vector of $L$'s eigenvalues. We write $\lambda^S = \prod_{i \in S} \lambda_i$, and denote by $\boldsymbol{\lambda}^{\wedge k}$ the $\binom{n}{k}$-vector $(\lambda^S)_{S \subseteq [n], |S|=k}$. Using Lemma 1, there exists $\boldsymbol{w} \in \mathbb{R}^{\binom{n}{k}}$ that sums to 1 such that

$$\frac{\det(L[S])^p}{\det(L^p[S])} = \frac{(\sum_{|S|=k} w_S \lambda^S)^p}{\sum_{|S|=k} w_S (\lambda^p)^S} = \left( \frac{M_1(\boldsymbol{w}; \boldsymbol{\lambda}^{\wedge k})}{M_p(\boldsymbol{w}; \boldsymbol{\lambda}^{\wedge k})} \right)^p$$
$$\leq r(p)^p.$$

Where the last inequality follows from Thm. 8. To lower bound $r_p(S)$, the same reasoning gives us $\frac{\det(L[S]^p)}{\det(L^p[S])} \geq 1$ and hence

$$r_p(S) \geq \frac{\det(I + L^p)}{Z_p} \geq \min_S \frac{\det L^p[S]}{(\det L[S])^p}$$
$$\geq \left( \frac{M_p(\boldsymbol{w'}; \boldsymbol{\lambda}^{\wedge k})}{M_1(\boldsymbol{w'}; \boldsymbol{\lambda}^{\wedge k})} \right)^p \geq r(p)^{-p},$$

where the second inequality follows $\frac{a+b}{c+d} \geq \min(\frac{a}{c}, \frac{b}{d})$. The same reasoning yields the result for $p > 1$.

Finally, some algebra shows that for fixed $p$, $r$ is an increasing function of $\kappa$, and so $r(\kappa_k, p)$ is upper bounded by $r(\kappa_{\lfloor n/2 \rfloor})$. $\square$

Figure 4: Evolution of the upper bound for $r(p, \kappa)$ from Thm. 7, which measures the $r$-closeness between the E-DPP with kernel $L$ and the DPP with kernel $L^p$.

# E  Mixing time as a function of ground set size for E-DPPs

(a) Proposal, $r = 2.0$      (b) Swapchain, $r = 2.0$

Figure 5: Influence of ground set size $n$ on mixing time

# F    Additional Nystrom sampling results

Figure 6: $L_2$ reconstruction error

Figure 7: Frobenius norm reconstruction error