[Reviews · NeurIPS 2018]

Reviewer 1



### Post-Rebuttal ### I have read the rebuttal and would like to thank you for the clarifications. ################## # Summary The paper introduces a concept of exponentiated strongly Rayleigh (ESR) measure motivated by the trade-offs between diversity and quality of items in a selected subset of a ground set with n items. A measure \mu over the subsets of a ground set of n items is called strongly Rayleigh (SR) if its generating polynomial P_{\mu}(z_1, ..., z_n) = \sum_S \mu(S) \prod_{i \in S} z_i is realstab.Theauthorstakethedefitionofareal stable' polynomial from [10] and define it as a polynomial without complex roots. An exponentiated strongly Rayleigh measure \mu is defined as a measure such that there exists a strongly Rayleigh measure \nu satisfying \mu(S) \prop \nu(S)^p for some p >= 0. Now, for p >1 the ESR measure promotes diversity whereas for p < 1 it allows less diverse items at the expense of their quality. For p=0 the ESR measure is uniform and for p=\infty it concentrates at the mode. # Comments/Details - The first theoretical result is Proposition 1 which states that there exists an ESR measure which is not a SR measure. This is a key result for establishing the `novelty' of ESR measures, showing that they are not contained in the class of SR measures. The proof of this proposition might need a clarification. In particular, if in the proof of this proposition one sets z=-l_{22}^p + u and w=-l_{11}^p + \overline{u} with u \in C such that || u ||^2 = l_{11}l_{22} - det(L^p) then one obtains a complex root of the polynomial p(z, w). In my understanding this then does not align with Definition 1. Moreover, the proof of Proposition 1 contains a number of claims without a proper mathematical derivation. If these results are from other papers, then those should be cited accordingly. Otherwise, a complete proof of the proposition is required. - The second theoretical result is Theorem 1, where it is shown that there exists a choice of p and a non-trivial matrix for which its exponent defines a determinantal point process that is an SR measure. This result is novel and establishes that some ESR measures remain SR. - To sample from an ESR measure the authors consider two algorithms. The first algorithm is an independent Metropolis--Hastings sampler with a proposal distribution given by a SR measure. It is quite strange that the authors do not cite the paper by Metropolis or some survey on MCMC methods because there is nothing novel about this algorithm. The second algorithm is pretty much the one published in [29]. I do not see any significant novelty in this part of the paper. - In order to study the mixing properties of the chains for sampling ESR measures the authors introduce a concept of closeness of measures (Definition 3) and provide a bound for the parameter defining it in Proposition 3. Following this, the main theoretical results are given which provide the mixing bounds for chains defined by Algorithms 1 and 2. Parts of these proofs can be obtained by mimicking previous derivations such as [5]. - In the concluding part of the section on ESR measures the authors focus on determinantal point processes and ESR measures defined by exponentiating a positive definite matrix defining such a process. For this specific case, the authors provide bounds on the parameter defining the closeness of distributions in mixing bounds for the considered sampling algorithms (see the comment on Definition 3). - The approach is evaluated empirically via three experiments: i) an experiment illustrating the mixing properties of the chain, ii) anomaly detection problem, and iii) Nystroem method with ESR landmarks (this experiment is replicated from the paper on DPP-Nystroem from [30]). The experiments illustrate the utility of the exponent in trade-offs between quality and diversity among items from a selected subset. ### Overall the main part of the paper is well written and easy to follow. In the appendix, it is quite difficult to follow the proof of Proposition 1 because of the claims that are expected to be accepted without a proof. In the summary of the contributions, it might be beneficial to be more explicit and state that the proposed algorithms are not novel contributions of the paper. The theoretical results providing mixing bounds for these samplers in combination with ESR measures are novel and interesting. Still the impression is that these are just minor adaptations of the previous work. I do, however, appreciate the flexibility introduced via the exponent p in ESR measures. It would be nice if the authors provided an ESR measure that is not in the context of determinantal point processes. The focus on determinantal point processes precludes other contributions and the utility of general ESR measures. ###

Reviewer 2



Strongly Rayleigh measure is distribution over the subsets of a ground set that balances quality and diversity. This paper introduces exponentiated strongly Rayleigh (ESR) that controls balance of diversity and quality. The main contributions of this work are to provide efficient ESR sampling algorithm using fast-mixing MCMC chains, which have been studied for SR measure in prior. The sampling algorithms requires either sampling from SR or swap a single element with proper probability. They analyze the algorithms with bound of mixing time of Markov chain, which is also upper-bounded by closeness between target ESR and a given SR. As a practical application, they study popular DPPs and dual volume sampling which have applied in numerous machine learning applications. Finally, authors evaluate their algorithms by reporting efficient mixing time. Furthermore, they apply ESR measure for outlier detection and kernel reconstruction tasks with comparable results. This paper proposes important works as authors first introduce exponentiated SR (ESR) measures, which resolves balancing diversity and quality of sampled subsets. And they first apply E-DPPs (popular classes of ESR) to outlier detection tasks and show reasonable results. In addition, this paper is very well written and easy to follow up. Overall I believe that this work is enough to accept to NIPS. Minor typos: The condition in Proposition 2 seems unnatural, i.e., mu \propto mu^p. ***** I have read the author feedback. I appreciate authors for providing the details about my questions. *****

Reviewer 3



The authors propose a generalisation of Strongly Raleigh discrete measures (SR) called Exponentiated Strongly Raleigh measures, which is an SR measured raised to a power p, where p>0 denotes the parameter of the exponentiation operation. Depending on the choice of p, many different cases are recovered of sample diversity vs sample quality. For instance, the discrete uniform, which is non SR, normal SR measures when p=1 as well as less and more sharper measures of diversity. As a by-product of this generalisation, the strength of negative dependence can be controlled by the tunning of this parameter, which is an interesting fact. The authors also derive an MCMC Metropolis-Hastings algorithm where the proposal is a Strongly Raleigh measure. This is the case because even though the ESR cannot be sampled from directly, the unnormalised version of it can be evaluated point wise, hence, one can evaluate the MH acceptance ratio. Another nice property of the proposed algorithm is that , when the proposal is sufficiently close to the target, the MCMC has fast mixing properties. In particular, in Proposition 2, a measure of closeness, called r-closeness, between this two measures is introduced. And in Section 3, bounds for this r-closeness are derived for the specific cases of k-DPPs and for dual volume sampling. There is also an empirical performance test of the mixing times in the experiments section. The experiments section show that this is a competitive approach in a determinental point process application for kernel reconstruction as well as an anomaly detection tasks. Overall, I believe that the paper is well written and the motivation is clear. There are noteworthy contributions in terms of theory and applications in ML. The authors have kindly provided feedback about the following specific comments/remarks and will incorporate them in the camera ready version which enhances the overall clarity of the submission: Line 26: what closure properties are you referring to? Please explain with more detail. Line 101: The equation below this line has “NA” instead of a number. Section 3: make clear what you mean by tractability, i.e. that it is possible to sample from SR measures as well as point wise evaluation of the unnormalised density. In contrast, it is not possible to sample from ESR but point wise evaluation is still possible.